# Clone MCMC: Parallel High-Dimensional Gaussian Gibbs Sampling

**Andrei-Cristian Bărbos**
IMS Laboratory
Univ. Bordeaux - CNRS - BINP
andbarbos@u-bordeaux.fr

**François Caron**
Department of Statistics
University of Oxford
caron@stats.ox.ac.uk

**Jean-François Giovannelli**
IMS Laboratory
Univ. Bordeaux - CNRS - BINP
giova@ims-bordeaux.fr

**Arnaud Doucet**
Department of Statistics
University of Oxford
doucet@stats.ox.ac.uk

## Abstract

We propose a generalized Gibbs sampler algorithm for obtaining samples approximately distributed from a high-dimensional Gaussian distribution. Similarly to Hogwild methods, our approach does not target the original Gaussian distribution of interest, but an approximation to it. Contrary to Hogwild methods, a single parameter allows us to trade bias for variance. We show empirically that our method is very flexible and performs well compared to Hogwild-type algorithms.

## 1 Introduction

Sampling high-dimensional distributions is notoriously difficult in the presence of strong dependence between the different components. The Gibbs sampler proposes a simple and generic approach, but may be slow to converge, due to its sequential nature. A number of recent papers have advocated the use of so-called "Hogwild Gibbs samplers", which perform conditional updates in parallel, without synchronizing the outputs. Although the corresponding algorithms do not target the correct distribution, this class of methods has shown to give interesting empirical results, in particular for Latent Dirichlet Allocation models [1, 2] and Gaussian distributions [3].

In this paper, we focus on the simulation of high-dimensional Gaussian distributions. In numerous applications, such as computer vision, satellite imagery, medical imaging, tomography or weather forecasting, simulation of high-dimensional Gaussians is needed for prediction, or as part of a Markov chain Monte Carlo (MCMC) algorithm. For example, [4] simulate high dimensional Gaussian random fields for prediction of hydrological and meteorological quantities. For posterior inference via MCMC in a hierarchical Bayesian model, elementary blocks of a Gibbs sampler often require to simulate high-dimensional Gaussian variables. In image processing, the typical number of variables (pixels/voxels) is of the order of $10^6/10^9$. Due to this large size, Cholesky factorization is not applicable; see for example [5] or [6].

In [7, 8] the sampling problem is recast as an optimisation one: a sample is obtained by minimising a perturbed quadratic criterion. The cost of the algorithm depends on the choice of the optimisation technique. Exact resolution is prohibitively expensive so an iterative solver with a truncated number of iterations is typically used [5] and the distribution of the samples one obtains is unknown.

In this paper, we propose an alternative class of iterative algorithms for approximately sampling high-dimensional Gaussian distributions. The class of algorithms we propose borrows ideas from optimization and linear solvers. Similarly to Hogwild algorithms, our sampler does not target the

distribution of interest but an approximation to this distribution. A single scalar parameter allows us to tune both the error and the convergence rate of the Markov chain, allowing to trade variance for bias. We show empirically that the method is very flexible and performs well compared to Hogwild algorithms. Its performance are illustrated on a large-scale image inpainting-deconvolution application.

The rest of the article is organized as follows. In Section 2, we review the matrix splitting techniques that have been used to propose novel algorithms to sample high-dimensional normals. In Section 3, we present our novel methodology. Section 4 provides the intuition for such a scheme, which we refer to as clone MCMC, and discusses some generalization of the idea to non-Gaussian target distributions. We compare empirically Hogwild and our methodology on a variety of simulated examples in Section 5. The application to image inpainting-deconvolution is developed in Section 6.

## 2 Background on matrix splitting and Hogwild Gaussian sampling

We consider a $d$-dimensional Gaussian random variable $X$ with mean $\mu$ and positive definite covariance matrix $\Sigma$. The probability density function of $X$, evaluated at $x = (x_1 \ldots, x_d)^\mathsf{T}$, is

$$\pi(x) \propto \exp \left\{ -\frac{1}{2} (x - \mu)^\mathsf{T} \Sigma^{-1} (x - \mu) \right\} \propto \exp \left\{ -\frac{1}{2} x^\mathsf{T} J x + h^\mathsf{T} x \right\}$$

where $J = \Sigma^{-1}$ is the precision matrix and $h = J\mu$ the potential vector. Typically, the pair $(h, J)$ is available, and the objective is to estimate $(\mu, \Sigma)$ or to simulate from $\pi$. For moderate-size or sparse precision matrices, the standard method for exact simulation from $\pi$ is based on the Cholesky decomposition of $\Sigma$, which has computational complexity $O(d^3)$ in the most general case [9]. If $d$ is very large, the cost of Cholesky decomposition becomes prohibitive and iterative methods are favoured due to their smaller cost per iteration and low memory requirements. A principled iterative approach to draw samples approximately distributed from $\pi$ is the single-site Gibbs sampler, which simulates a Markov chain $(X^{(i)})_{i=1,2,\ldots}$ with stationary distribution $\pi$ by updating each variable in turn from its conditional distribution. A complete update of the $d$ variables can be written in matrix form as

$$X^{(i+1)} = -(D + L)^{-1} L^\mathsf{T} X^{(i)} + (D + L)^{-1} Z^{(i+1)}, \quad Z^{(i+1)} \sim \mathcal{N}(h, D) \tag{1}$$

where $D$ is the diagonal part of $J$ and $L$ is is the strictly lower triangular part of $J$. Equation (1) highlights the connection between the Gibbs sampler and linear iterative solvers as

$$\mathbb{E}[X^{(i+1)} | X^{(i)} = x] = -(D + L)^{-1} L^\mathsf{T} x + (D + L)^{-1} h$$

is the expression of the Gauss-Seidel linear iterative solver update to solve the system $J\mu = h$ for a given pair $(h, J)$. The single-site Gaussian Gibbs sampler can therefore be interpreted as a stochastic version of the Gauss-Seidel linear solver. This connection has been noted by [10] and [11], and later exploited by [3] to analyse the Hogwild Gibbs sampler and by [6] to derive a family of Gaussian Gibbs samplers.

The Gauss-Seidel iterative solver is just a particular example of a larger class of matrix splitting solvers [12]. In general, consider the linear system $J\mu = h$ and the matrix splitting $J = M - N$, where $M$ is invertible. Gauss-Seidel corresponds to setting $M = D + L$ and $N = -L^\mathsf{T}$. More generally, [6] established that the Markov chain with transition

$$X^{(i+1)} = M^{-1} N X^{(i)} + M^{-1} Z^{(i+1)}, \quad Z^{(i+1)} \sim \mathcal{N}(h, M^\mathsf{T} + N) \tag{2}$$

admits $\pi$ as stationary distribution if and only if the associated iterative solver with update

$$x^{(i+1)} = M^{-1} N x^{(i)} + M^{-1} h$$

is convergent; that is if and only if $\rho(M^{-1}N) < 1$, where $\rho$ denotes the spectral radius. Using this result, [6] built on the large literature on linear iterative solvers in order to derive generalized Gibbs samplers with the correct Gaussian target distribution, extending the approaches proposed by [10, 11, 13].

The practicality of the iterative samplers with transition (2) and matrix splitting $(M, N)$ depends on

- How easy it is to solve the system $Mx = r$ for any $r$,

- How easy it is to sample from $\mathcal{N}(0, M^{\mathsf{T}} + N)$.

As noted by [6], there is a necessary trade-off here. The Jacobi splitting $M = D$ would lead to a simple solution to the linear system, but sampling from a Gaussian distribution with covariance matrix $M^{\mathsf{T}} + N$ would be as complicated as solving the original sampling problem. The Gauss-Seidel splitting $M = D + L$ provides an interesting trade-off as $Mx = r$ can be solved by substitution and $M^{\mathsf{T}} + N = D$ is diagonal. The method of successive over-relaxation (SOR) uses a splitting $M = \omega^{-1}D + L$ with an additional tuning parameter $\omega > 0$. In both the SOR and Gauss-Seidel cases, the system $Mx = r$ can be solved by substitution in $O(d^2)$, but the resolution of the linear system cannot be parallelized.

All the methods discussed so far asymptotically sample from the correct target distribution. The Hogwild Gaussian Gibbs sampler does not, but its properties can also be analyzed using techniques from the linear iterative solver literature as demonstrated by [3]. For simplicity of exposure, we focus here on the Hogwild sampler with blocks of size 1. In this case, the Hogwild algorithm simulates a Markov chain with transition

$$X^{(i+1)} = M_{\text{Hog}}^{-1} N_{\text{Hog}} X^{(i)} + M_{\text{Hog}}^{-1} Z^{(i+1)}, \quad Z^{(i+1)} \sim \mathcal{N}(h, M_{\text{Hog}})$$

where $M_{\text{Hog}} = D$ and $N_{\text{Hog}} = -(L + L^{\mathsf{T}})$. This update is highly amenable to parallelization as $M_{\text{Hog}}$ is diagonal thus one can easily solve the system $M_{\text{Hog}} x = r$ and sample from $\mathcal{N}(0, M_{\text{Hog}})$. [3] showed that if $\rho(M_{\text{Hog}}^{-1} N_{\text{Hog}}) < 1$, the Markov chain admits $\mathcal{N}(\mu, \widetilde{\Sigma})$ as stationary distribution where

$$\widetilde{\Sigma} = (I + M_{\text{Hog}}^{-1} N_{\text{Hog}})^{-1} \Sigma.$$

The above approach can be generalized to blocks of larger sizes. However, beyond the block size, the Hogwild sampler does not have any tunable parameter allowing us to modify its incorrect stationary distribution. Depending on the computational budget, we may want to trade bias for variance. In the next Section, we describe our approach, which offers such flexibility.

## 3 High-dimensional Gaussian sampling

Let $J = M - N$ be a matrix splitting, with $M$ positive semi-definite. Consider the Markov chain $(X^{(i)})_{i=1,2,...}$ with initial state $X^{(0)}$ and transition

$$X^{(i+1)} = M^{-1} N X^{(i)} + M^{-1} Z^{(i+1)}, \quad Z^{(i+1)} \sim \mathcal{N}(h, 2M). \tag{3}$$

The following theorem shows that, if the corresponding iterative solver converges, the Markov chain converges to a Gaussian distribution with the correct mean and an approximate covariance matrix.

**Theorem 1.** *If $\rho(M^{-1}N) < 1$, the Markov chain $(X^{(i)})_{i=1,2,...}$ defined by (3) has stationary distribution $\mathcal{N}(\mu, \widetilde{\Sigma})$ where*

$$\widetilde{\Sigma} = 2\left(I + M^{-1}N\right)^{-1}\Sigma$$
$$= (I - \frac{1}{2}M^{-1}\Sigma^{-1})^{-1}\Sigma.$$

*Proof.* The equivalence between the convergence of the iterative linear solvers and their stochastic counterparts was established in [6, Theorem 1]. The mean $\widetilde{\mu}$ of the stationary distribution verifies the recurrence

$$\widetilde{\mu} = M^{-1}N\widetilde{\mu} + M^{-1}\Sigma^{-1}\mu$$

hence

$$(I - M^{-1}N)\widetilde{\mu} = M^{-1}\Sigma^{-1}\mu \quad \Leftrightarrow \quad \widetilde{\mu} = \mu$$

as $\Sigma^{-1} = M - N$. For the covariance matrix, consider the $2d$-dimensional random variable

$$\begin{pmatrix} Y_1 \\ Y_2 \end{pmatrix} = \mathcal{N}\left(\begin{pmatrix} \mu \\ \mu \end{pmatrix}, \begin{pmatrix} M/2 & -N/2 \\ -N/2 & M/2 \end{pmatrix}^{-1}\right) \tag{4}$$

Then using standard manipulations of multivariate Gaussians and the inversion lemma on block matrices we obtain

$$Y_1|Y_2 \sim \mathcal{N}(M^{-1}NY_2 + M^{-1}h, 2M^{-1})$$
$$Y_2|Y_1 \sim \mathcal{N}(M^{-1}NY_1 + M^{-1}h, 2M^{-1})$$

and

$$Y_1 \sim \mathcal{N}(\mu, \widetilde{\Sigma}), \quad Y_2 \sim \mathcal{N}(\mu, \widetilde{\Sigma})$$

□

The above proof is not constructive, and we give in Section 4 the intuition behind the choice of the transition and the name clone MCMC.

We will focus here on the following matrix splitting

$$M = D + 2\eta I, \quad N = 2\eta I - L - L^\mathsf{T} \tag{5}$$

where $\eta \geq 0$. Under this matrix splitting, $M$ is a diagonal matrix and an iteration only involves a matrix-vector multiplication of computational cost $O(d^2)$. This operation can be easily parallelized. Each update has thus the same computational complexity as the Hogwild algorithm. We have

$$\widetilde{\Sigma} = (I - \frac{1}{2}(D + 2\eta I)^{-1}\Sigma^{-1})^{-1}\Sigma.$$

Since $M^{-1} \to 0$ and $M^{-1}N \to I$ for $\eta \to \infty$, we have

$$\lim_{\eta \to \infty} \widetilde{\Sigma} = \Sigma, \quad \lim_{\eta \to \infty} \rho(M^{-1}N) = 1.$$

The parameter $\eta$ is an easily interpretable tuning parameter for the method: as $\eta$ increases, the stationary distribution of the Markov chain becomes closer to the target distribution, but the samples become more correlated.

For example, consider the target precision matrix $J = \Sigma^{-1}$ with $J_{ii} = 1$, $J_{ij} = -1/(d+1)$ for $i \neq j$ and $d = 1000$. The proposed sampler is run for different values of $\eta$ in order to estimate the covariance matrix $\Sigma$. Let $\hat{\Sigma} = 1/n_s \sum_{i=1}^{n_s}(X^{(i)} - \hat{\mu})^\mathsf{T}(X^{(i)} - \hat{\mu})$ be the estimated covariance matrix where $\hat{\mu} = 1/n_s \sum_{i=1}^{n_s} X^{(i)}$ is the estimated mean. The Figure 1(a) reports the bias term $||\Sigma - \widetilde{\Sigma}||$, the variance term $||\hat{\Sigma} - \widetilde{\Sigma}||$ and the overall error $||\Sigma - \hat{\Sigma}||$ as a function of $\eta$, using $n_s = 10000$ samples and 100 replications, with $|| \cdot ||$ the $\ell_2$ (Frobenius) norm. As $\eta$ increases, the bias term decreases while the variance term increases, yielding an optimal value at $\eta \simeq 10$. Figure 1(b-c) show the estimation error for the mean and covariance matrix as a function of $\eta$, for different sample sizes. Figure 2 shows the estimation error as a function of the sample size for different values of $\eta$.

The following theorem gives a sufficient condition for the Markov chain to converge for any value $\eta$.

**Theorem 2.** *Let $M = D + 2\eta I$, $N = 2\eta I - L - L^\mathsf{T}$. A sufficient condition for $\rho(M^{-1}N) < 1$ for all $\eta \geq 0$ is that $\Sigma^{-1}$ is strictly diagonally dominant.*

*Proof.* $M$ is non singular, hence

$$\det(M^{-1}N - \lambda I) = 0 \Leftrightarrow \det(N - \lambda M) = 0.$$

$\Sigma^{-1} = M - N$ is diagonally dominant, hence $\lambda M - N = (\lambda - 1)M + M - N$ is also diagonally dominant for any $\lambda \geq 1$. From Gershgorin's theorem, a diagonally dominant matrix is nonsingular, so $\det(N - \lambda M) \neq 0$ for all $\lambda \geq 1$. We conclude that $\rho(M^{-1}N) < 1$. □

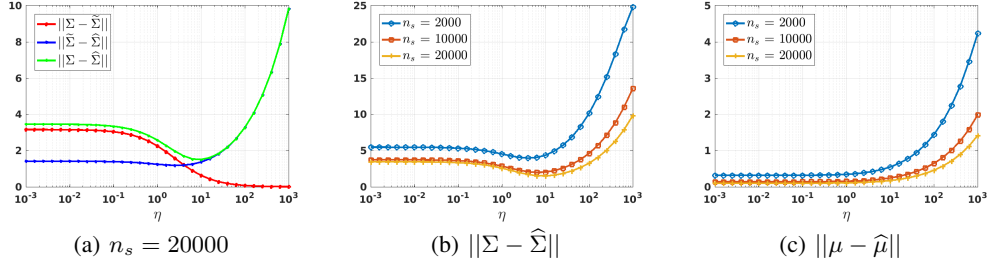

(a) $n_s = 20000$          (b) $||\Sigma - \widehat{\Sigma}||$          (c) $||\mu - \widehat{\mu}||$

Figure 1: Influence of the tuning parameter $\eta$ on the estimation error

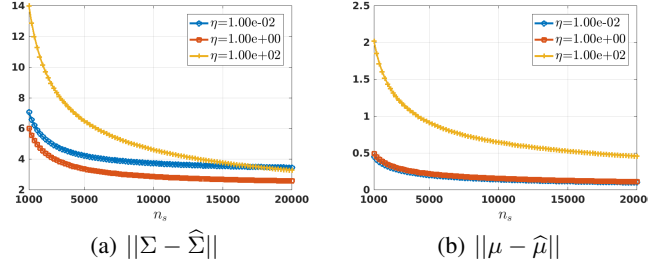

(a) $||\Sigma - \widehat{\Sigma}||$          (b) $||\mu - \widehat{\mu}||$

Figure 2: Influence of the sample size on the estimation error

## 4   Clone MCMC

We now provide some intuition on the construction given in Section 3, and justify the name given to the method. The joint pdf of $(Y_1, Y_2)$ on $\mathbb{R}^{2d}$ defined in (4) with matrix splitting (5) can be expressed as

$$\widetilde{\pi}_\eta(y_1, y_2) \propto \exp\{-\frac{\eta}{2}(y_1 - y_2)^\mathsf{T}(y_1 - y_2)\}$$

$$\times \exp\{-\frac{1}{4}(y_1 - \mu)^\mathsf{T} D(y_1 - \mu) - \frac{1}{4}(y_1 - \mu)^\mathsf{T}(L + L^\mathsf{T})(y_2 - \mu)\}$$

$$\times \exp\{-\frac{1}{4}(y_2 - \mu)^\mathsf{T} D(y_2 - \mu) - \frac{1}{4}(y_2 - \mu)^\mathsf{T}(L + L^\mathsf{T})(y_1 - \mu)\}$$

We can interpret the joint pdf above as having cloned the original random variable $X$ into two dependent random variables $Y_1$ and $Y_2$. The parameter $\eta$ tunes the correlation between the two variables, and $\widetilde{\pi}_\eta(y_1|y_2) = \prod_{k=1}^d \widetilde{\pi}_\eta(y_{1k}|y_2)$, which allows for straightforward parallelization of the method. As $\eta \to \infty$, the clones become more and more correlated, with $corr(Y_1, Y_2) \to 1$ and $\widetilde{\pi}_\eta(y_1) \to \pi(y_1)$.

The idea can be generalized further to pairwise Markov random fields. Consider the target distribution

$$\pi(x) \propto \exp\left(-\sum_{1 \leq i \leq j \leq d} \psi_{ij}(x_i, x_j)\right)$$

for some potential functions $\psi_{ij}$, $1 \leq i \leq j \leq d$. The clone pdf is

$$\widetilde{\pi}(y_1, y_2) \propto \exp\{-\frac{\eta}{2}(y_1 - y_2)^\mathsf{T}(y_1 - y_2) - \frac{1}{2} \sum_{1 \leq i \leq j \leq d} (\psi_{ij}(y_{1i}, y_{2i}) + \psi_{ij}(y_{2i}, y_{1i}))\}$$

where

$$\widetilde{\pi}(y_1|y_2) = \prod_{k=1}^d \widetilde{\pi}(y_{1k}|y_2).$$

Assuming $\widetilde{\pi}$ is a proper pdf, we have $\widetilde{\pi}(y_1) \to \pi(y_1)$ as $\eta \to \infty$.

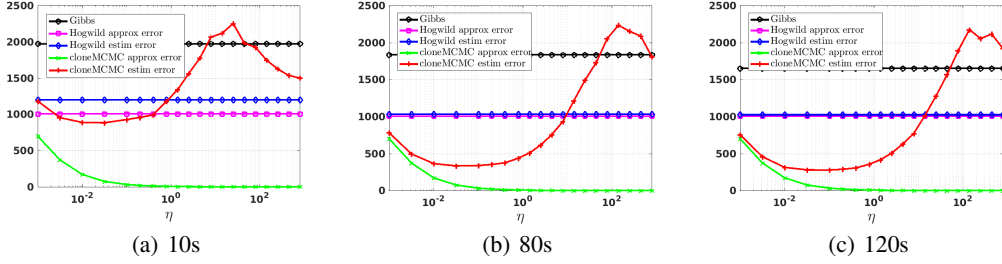

Figure 3: Estimation error for the covariance matrix $\Sigma_1$ for fixed computation time, $d = 1000$.

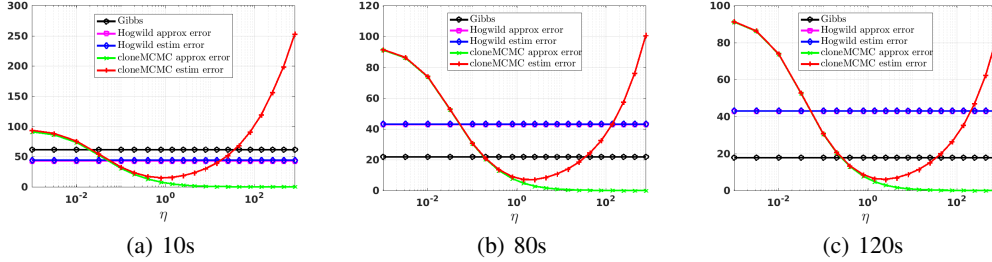

Figure 4: Estimation error for the covariance matrix $\Sigma_2$ for fixed computation time, $d = 1000$.

## 5  Comparison with Hogwild and Gibbs sampling

In this section, we provide an empirical comparison of the proposed approach with the Gibbs sampler and Hogwild algorithm, using the splitting (5). Note that in order to provide a fair comparison between the algorithms, we only consider the single-site Gibbs sampling and block-1 Hogwild algorithms, whose updates are respectively given in Equations (1) and (2). Versions of all three algorithms could also be developed with blocks of larger sizes.

We consider the following two precision matrices.

$$\Sigma_1^{-1} = \begin{pmatrix} 1 & -\alpha & & & \\ -\alpha & 1+\alpha^2 & -\alpha & & \\ & \ddots & \ddots & \ddots & \\ & & -\alpha & 1+\alpha^2 & -\alpha \\ & & & -\alpha & 1 \end{pmatrix}, \quad \Sigma_2^{-1} = \begin{pmatrix} \ddots & \ddots & \ddots & \ddots & \ddots & \\ & 0.15 & 0.3 & 1 & 0.3 & 0.15 \\ & & \ddots & \ddots & \ddots & \ddots & \ddots \end{pmatrix}$$

where for the first precision matrix we have $\alpha = 0.95$. Experiments are run on GPU with 2688 CUDA cores. In order to compare the algorithms, we run each algorithm for a fixed execution time (10s, 80s and 120s). Computation time per iteration for Hogwild and Clone MCMC are similar, and they return a similar number of samples. The computation time per iteration of the Gibbs sampling is much higher, due to the lack of parallelization, and it returns less samples. For Hogwild and Clone MCMC, we report both the approximation error $||\Sigma - \widetilde{\Sigma}||$ and the estimation error $||\Sigma - \widehat{\Sigma}||$. For Gibbs, only the estimation error is reported.

Figures 3 and 4 show that, for a range of values of $\eta$, our method outperforms both Hogwild and Gibbs, whatever the execution time. As the computational budget increases, the optimal value for $\eta$ increases.

## 6  Application to image inpainting-deconvolution

In order to demonstrate the usefulness of the approach, we consider an application to image inpainting-deconvolution. Let

$$Y = THX + B, \ \ B \sim \mathcal{N}(0, \Sigma_b) \tag{6}$$

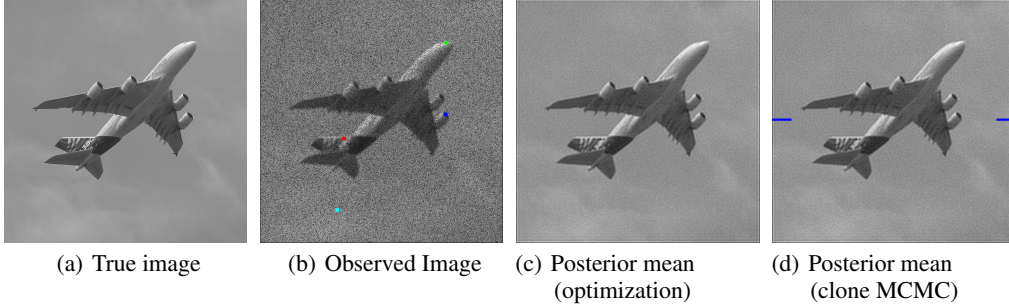

| (a) True image | (b) Observed Image | (c) Posterior mean (optimization) | (d) Posterior mean (clone MCMC) |

Figure 5: Deconvolution-Interpolation results

be the observation model where $Y \in \mathbb{R}^n$ is the observed image, $X \in \mathbb{R}^d$ is the true image, $B \in \mathbb{R}^n$ is the noise component, $H \in \mathbb{R}^{d \times d}$ is the convolution matrix and $T \in \mathbb{R}^{n \times d}$ is the truncation matrix. The observation noise is assumed to be independent of $X$ with $\Sigma_b^{-1} = \gamma_b I$ and $\gamma_b = 10^{-2}$. Assume

$$X \sim \mathcal{N}(0, \Sigma_x)$$

with

$$\Sigma_x^{-1} = \gamma_0 1_d 1_d^{\mathsf{T}} + \gamma_1 C C^{\mathsf{T}}$$

wherein $1_d$ is a column vector of size $d$ having all elements equal to $1/d$, $C$ is the block-Toeplitz convolution matrix corresponding to the 2D Laplacian filter and $\gamma_0 = \gamma_1 = 10^{-2}$.

The objective is to sample from the posterior distribution

$$X|Y = y \sim \mathcal{N}(\mu_{x|y}, \Sigma_{x|y})$$

where

$$\Sigma_{x|y}^{-1} = H^{\mathsf{T}} T^{\mathsf{T}} \Sigma_b^{-1} T H + \Sigma_x^{-1}$$
$$\mu_{x|y} = \Sigma_{x|y} H^{\mathsf{T}} T^{\mathsf{T}} \Sigma_b^{-1} y.$$

The true unobserved image is of size $1000 \times 1000$, hence the posterior distribution corresponds to a random variable of size $d = 10^6$. We have considered that $20\%$ of the pixels are not observed. The true image is given in Figure 5(a); the observed image is given in Figure 5(b).

In this high-dimensional setting with $d = 10^6$, direct sampling via Cholesky decomposition or standard single-site Gibbs algorithm are not applicable. We have implemented the block-1 Hogwild algorithm. However, in this scenario the algorithm diverges, which is certainly due to the fact that the spectral radius of $M_{\text{Hog}}^{-1} N_{\text{Hog}}$ is greater than 1.

We run our clone MCMC algorithm for $n_s = 19000$ samples, out of which the first 4000 were discarded as burn-in samples, using as initialization the observed image, with missing entries padded with zero. The tuning parameter $\eta$ is set to 1. Figure 5(c) contains the reconstructed image that was obtained by numerically maximizing the posterior distribution using gradient ascent. We shall take this image as reference when evaluating the reconstructed image computed as the posterior mean from the drawn samples. The reconstructed image is given in Figure 5(d).

If we compare the restored image with the one obtained by the optimization approach we can immediately see that the two images are visually very similar. This observation is further reinforced by the top plot from Figure 6 where we have depicted the same line of pixels from both images. The line of pixels that is displayed is indicated by the blue line segments in Figure 5(d). The traces in grey represent the $99\%$ credible intervals. We can see that for most of the pixels, if not for all for that matter, the estimated value lies well within the $99\%$ credible intervals. The bottom plot from Figure 6 displays the estimated image together with the true image for the same line of pixels, showing an accurate estimation of the true image. Figure 7 shows traces of the Markov chains for 4 selected pixels. Their exact position is indicated in Figure 5(b). The red marker corresponds to an observed pixel from a region having a mid-grey tone. The green marker corresponds to an observed pixel from a white tone region. The dark blue marker corresponds to an observed pixel from dark tone region.

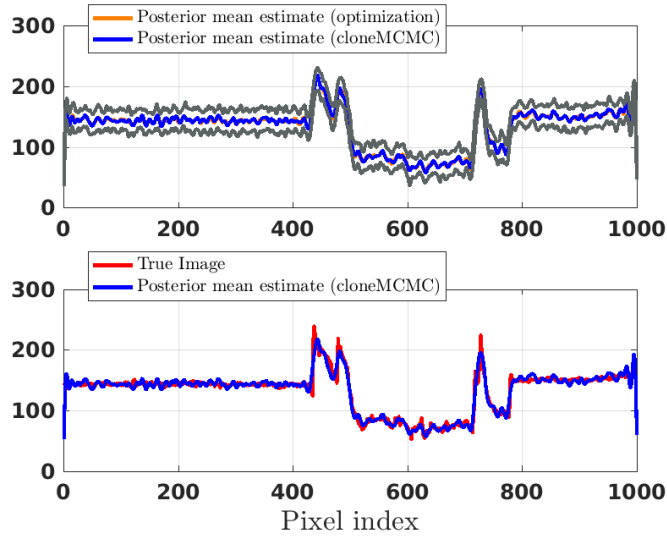

Figure 6: Line of pixels from the restored image

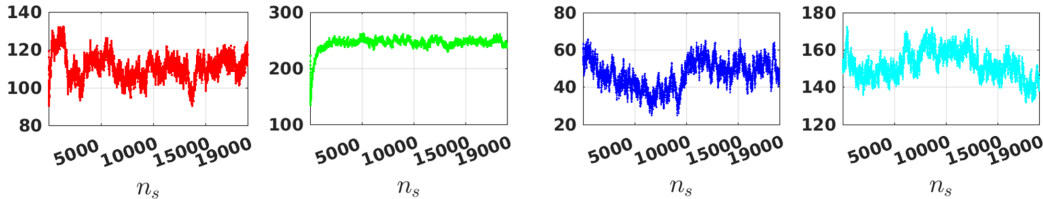

Figure 7: Markov chains for selected pixels, clone MCMC

The cyan marker corresponds to an observed pixel from a region having a tone between mid-grey and white.

The choice of $\eta$ can be a sensible issue for the practical implementation of the algorithm. We observed empirically convergence of our algorithm for any value $\eta$ greater than 0.075. This is a clear advantage over Hogwild, as our approach is applicable in settings where Hogwild is not as it diverges, and offers an interesting way of controlling the bias/variance trade-off. We plan to investigate methods to automatically choose the tuning parameter $\eta$ in future work.

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
