[Reviews · NeurIPS 2017]

Reviewer 1



The paper introduce a novel MCMC method for drawing from high dimensional Gaussian distributions. Clearly, there already exist direct methods for sampling a Gaussian density (involving Cholesky decomposition). The authors affirm that in an high dimensional framework is better to use of the proposed MCMC. - Is there, in the literature and applications, a clear demand for sampling high dimensional Gaussians (where Cholesky fails, for instance)? I believe few cases. - In these cases, perhaps a Sigma Points-type approach could be used avoiding any numerical problem (it is a deterministic approach but is able to summarize the Gaussian moments). Please, discuss.

Reviewer 2



This paper proposes a new parallel approximate sampler for high-dimensional Gaussian distributions. The algorithm is a special case of a larger class of iterative samplers based on a transition equation (2) and matrix splitting that is analysed in [9]. The algorithm is similar to the Hogwild sampler in term of the update formula and the way of bias analysing, but it is more flexible in the sense that there is a scalar parameter to trade-off the bias and variance of the proposed sampler. I appreciate the detailed introduction about the mathematical background of the family of sampling algorithms and related works. It is also easy to follow the paper and understand the merit of the proposed algorithm. The illustration of the decomposition of the variance and bias in Figure 1 gives a clear explanation about the role of \eta. Experiments on two covariance matrix shows that the proposed algorithm achieves lower error than Hogwild for some range of the \eta value. However, the range changes with the computation budget as well as the value of the covariance matrix. It is not clear to me if the proposed algorithm is better than Hogwild in practice if we do not have a good method/heuristic to choose a proper value for \eta. Another problem with the experiment evaluation is that there is no comparison in the only real applications, image pinpointing-deconvolution. Does the clone MCMC give a better image recovery than other competing methods?

Reviewer 3



The authors propose a novel Gibbs sampling approach for approximately drawing from high dimensional Gaussian distributions. I very much enjoyed reading this paper, which has wide applicability. I thought it was nicely written, both the introduction and the explanation of the proposed method were very clear, and the empirical results show clear benefits over existing Hogwild-type algorithms. My only comment would be for the authors to provide some more practical information regarding the choice of the tuning parameter - how might this be chosen for particular applications? Is there some way to learn an appropriate bias/variance trade-off in real time?